# Determination of low-level temperature profiles from microwave radiometer observations during rain

Andreas Foth[1], Moritz Lochmann[1], Pablo Saavedra Garfias[1], and Heike Kalesse-Los[1]

[1]Leipzig Institute for Meteorology, Leipzig University, Leipzig, Germany

**Correspondence:** Andreas Foth (andreas.foth@uni-leipzig.de)

**Abstract.** Usually, microwave radiometer observations have to be discarded during rain. The radomes of the receiver antenna get wet which hampers accurate measurements since the retrieval algorithms to derive atmospheric quantities are not trained for rain events. The reason for the latter is, that the rain drops dominate the microwave signal compared to the weaker signal from atmospheric gases. To account for this, radiative transfer simulations need to include the electromagnetic properties of rain, which usually requires more complicated and expensive simulations. In this work, the performance of newly developed microwave radiometer retrievals that are not based on rain simulations is evaluated to assess how they work during rain events. It is shown that it is possible to retrieve low-level temperature profiles during rain by omitting certain frequencies and zenith observations. Retrievals with various combinations of elevation angles and frequencies are evaluated. It is presented that, retrievals based on scanning mode observations with angles below 30° without zenith observation and only the lesser transparent upper four HATPRO microwave radiometer frequencies of the V-band (54.94, 56.66, 57.3, 58 GHz) provides the best results. An analysis of the calculated degrees of freedom of the signal shows that the retrieval of temperature profiles up to 3 km for no rain, 1.5 km for light to moderate rain and 1. km for very heavy rain is driven by the HATPRO observation and not by climatology. Finally, the performance of the temperature profile retrieval is explained using a case study in Lindenberg, Germany, and evaluated with temperature profiles from European Center for Medium-range Weather Forecasts (ECMWF) model for different rainfall intensities. The results show that the higher the rainfall rate, the larger the deviation of the retrieved microwave radiometer temperature profile from the ECMWF model output. The proposed retrievals for temperature profiles up to at least 1.5 km for rain rates below 0.5 and below $2.5\,\mathrm{mm\,h^{-1}}$ have uncertainties of less than 1 and 2 K, respectively, compared to ECMWF model output profiles.

## 1 Introduction

The continuous development and improvement of weather and climate models poses a great challenge to atmospheric remote sensing. For the evaluation of the models, increasingly better-resolved measurements and retrieval methods are needed, e.g. regarding air temperature profiles. Conventional remote sensing observational approaches mainly fail as they are incapable to provide continuous observations of temperature profiles under all weather conditions and especially during rain. Snow and ice clouds do not emit in the considered spectrum, hence they are not taken into account here. Ground-based Raman lidars can usually measure temperature and humidity profiles only below clouds and certainly not during rain (Wandinger,

2005). Radiosondes can provide these atmospheric profiles with high vertical resolution, but they are only routinely available at selected locations and at maximum every 6 hours. Additionally, radiosondes show a significant sonde-to-sonde variability (Nash et al., 2005) as well as a dry bias (Turner et al., 2003).

Multifrequency microwave radiometers (MWR) can provide temporally highly resolved profiles of temperature and humid-
ity, as well as integrated water vapor and liquid water path (Solheim et al., 1998; Güldner and Spänkuch, 1999; Westwater et al., 2005; Rose et al., 2005). Measurements at different elevation angles increase the accuracy of the derived temperature profiles in the atmospheric boundary layer (Crewell and Löhnert, 2007). The measurement uncertainties are described by Böck et al. (2024). Valid retrievals are, however, generally only possible during non-raining conditions (Ware et al., 2004). During rain the atmosphere becomes opaque at high frequencies of the V-band (54.94, 56.66, 57.3, 58 GHz) and no information can be
retrieved from higher altitudes. Additionally, the instrument gets wet and the received signal is dominated by the liquid water accumulated on the instrument. In previous studies Cimini et al. (2011) and Ware et al. (2013) compared retrieved profiles of temperature and absolute humidity from a neural network approach (scanning and zenith) and a one-dimensional variational (1DVAR) technique under $15°$ elevation angle with soundings during all weather conditions. For atmospheric profiling from the surface to $10\,km$, Cimini et al. (2011) obtained retrieval errors within $1.5\,K$ for temperature and $0.5\,g\,m^{-3}$ for absolute
humidity. Xu et al. (2014) retrieved thermodynamic profiles such as temperature and humidity as well as liquid water profiles by using off-zenith MWR observations at $15°$ elevation to reduce the impact of rain on the measurements using a neural net-
work approach. The temperature bias and root mean square error against radiosondes in precipitation were reduced from 3.6 and 4.2 K to 1.3 and 3.1 K, respectively, compared to the zenith MWR observations. Later, Araki et al. (2015) compared the method from Xu et al. (2014) with 1DVAR technique using zenith and off-zenith observation during raining and non-raining
conditions. Their results were evaluated with co-located radiosondes and they showed that the error in retrieved temperature and water vapor profiles in the low-level troposphere can be reduced by the 1DVAR technique even during rainfall with rain rates less than $1\,mm\,h^{-1}$ by using off-zenith observations. In the presented study, the impact of rain is reduced by using eleva-
tion scans only of off-zenith measurements, i.e., at lower elevation angles, because liquid water usually accumulates at the top of the MWR. Furthermore, the influence of rain can be reduced by using only the higher frequencies of the oxygen absorption
complex (V-band) in which the signals are almost saturated and will thus not be influenced so strongly by liquid water. The method presented here can be applied to standard measurement modes and does not require any changes in measurement setup. We show that there is no need to constantly change the measurement mode according to the weather conditions.

The structure of the manuscript is as follows: used instruments such as MWR and radiosondes, European Center for Medium-
range Weather Forecasts (ECMWF) model, ERA5 model and radiative transfer models are introduced in the Sec. 2 followed
by a description of the retrieval methodology in Sec. 3. The retrieval performance based on simulations and observations as well as a comparison of the observations with the ECMWF model output are evaluated in Sec. 4.

## 2 Instrument and Models

Almost all remote sensing data presented in this work were gathered at the Meteorological Observatory Lindenberg - Richard-Assmann-Observatory (MOL-RAO, 52.208°N, 14.118°E) in Lindenberg, Germany, during an instrument intercomparison campaign from July 16, 2020 until October 10, 2020. In addition to that, MWR data presented in Sec. 3 was gathered at the Leipzig Institute for Meteorology, Leipzig University (51.333°N, 12.389°E). The used instruments and models are explained in the following subsections.

### 2.1 Microwave radiometer HATPRO

The humidity and temperature profiler (HATPRO, generation 5) is a fully automatic microwave radiometer (MWR) from the manufacturer Radiometer Physics GmbH (Rose et al., 2005). It is a passive instrument and measures atmospheric emission at 14 frequencies along the microwave spectrum with a high temporal resolution in the order of seconds. Seven frequencies are situated along the upper wing of the water vapor absorption band at 22 GHz (K-band) and seven at the lower wing of the oxygen absorption complex at 58 GHz (V-band). For both absorption bands, HATPRO has its own antenna, which measured signal is converted into voltages at the individual frequencies. The voltages are then calibrated to brightness temperatures by automated calibrations (Kazama et al., 1999; Maschwitz et al., 2013; Küchler et al., 2016). The antennae are situated below a radome sheet, which is transparent in the microwave region. It is made of foam with a hydrophobic coating. HATPRO utilizes a rain mitigation system which blows a constant strong air stream over the radome. Nevertheless, during heavy or prolonged rainfall, liquid water might still accumulate on the radome's top, especially if the radome has aged, as is the case during long-term use in the field. An aged radome with a weathered coating absorbs moisture like a sponge. This prevents the accurate determination of atmospheric variables during rain.

In order to estimate column-integrated variables such as the integrated water vapor and liquid water path, as well as vertical profiles of temperature and humidity, so-called retrievals must be created (Löhnert and Crewell, 2003). Retrievals are based on artificial neural networks or multi-linear regression models which are trained on relations between measured brightness temperatures and the wanted quantity from radiosondes or numerical weather prediction model output. Observations under different elevation angles enhance the accuracy of the retrieved temperature profile within the atmospheric boundary layer (Crewell and Löhnert, 2007). A sketch showing the HATPRO measurements at default elevation angles color-coded by zenith and off-zenith is illustrated in Fig. 1. Those angles were intentionally selected to represent 1, 2, 3, 4, 5, 7, 9, 11, 12, and 14 air masses.

### 2.2 Radiosondes

Radiosondes provide highly resolved vertical information of atmospheric temperature, humidity and pressure. Here we used a large data set of 10172 Vaisala RS41 soundings from January 2015 to April 2024. This serves as input into radiative transfer calculations to create the synthetic brightness temperatures used for the retrieval algorithm to estimate temperature profiles

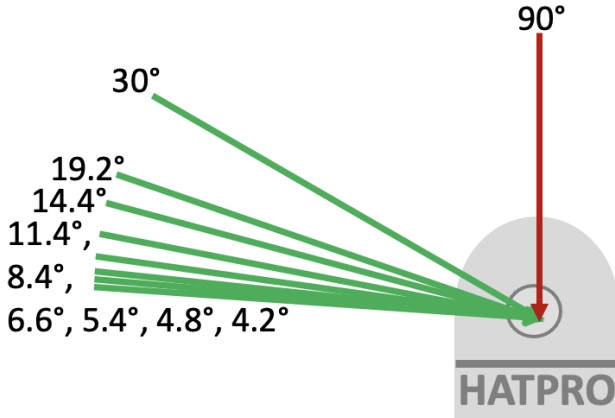

**Figure 1.** HATPRO's default set of elevation angles. Green and red arrows show off-zenith and zenith elevation angles, respectively.

(see Sec. 2.5). For the comparisons of temperature profiles in Sec. 4.2, Vaisala RS41 radiosondes are used, too (Jensen et al., 2016; Sun et al., 2019). In the presented work, all radiosondes were launched at MOL-RAO.

### 2.3 European Center for Medium-range Weather Forecasts model

In this study, temperature profiles from ECMWF Integrated Forecast System (IFS) are used to evaluate the retrieved temperature profiles from the MWR observations. This is done because the ECMWF-IFS model data is available in a higher temporal resolution (hourly) than that of the radiosondes. The model data used here are stored in the Cloudnet categorization product (Illingworth et al., 2007) which is freely available at https://cloudnet.fmi.fi/search/data?site=lindenberg (last access, 3 Sep, 2024).

### 2.4 ERA5

ERA5 (ECMWF Reanalysis v5) is the fifth generation of ECMWF's atmospheric reanalysis of global climate (Hersbach et al., 2020). ERA5 is produced by the Copernicus Climate Change Service (C3S) at ECMWF and covers data from 1940 to present. Here, hourly profiles of temperature, humidity, pressure and cloud liquid with a vertical resolution of 137 pressure levels from the surface up to a height of 80 km are extracted from the global data-set for the MOL-RAO site. 173 088 profiles from the ERA5 data set from 2004 to 2023 are used as input for the radiative transfer calculation for the temperature retrieval creation.

### 2.5 Non-scattering microwave radiative transfer model

Based on Simmer (1994), the non-scattering microwave radiative transfer is applied to calculate the brightness temperatures of each profile from 10 172 radio-soundings and 173 088 ERA5 profiles. This results in a data-set of 183 260 profiles with corresponding calculated brightness temperatures which serve as base for the retrieval generation. It uses the 2022 Rosenkranz gas absorption (Larosa et al., 2024) and liquid water absorption by Liebe (Liebe et al., 1993). The Rosenkranz gas absorption

model is corrected for the water vapor continuum absorption according to Turner et al. (2009). Uncertainty of atmospheric microwave absorption models and their impact on ground-based radiometer simulations and retrievals are extensively described in Cimini et al. (2018). The model code is written in the interactive data language (idl) and was ,e.g. also applied in Löhnert and Crewell (2003); Löhnert et al. (2007); Foth and Pospichal (2017).

## 2.6 Passive and Active Microwave TRAnsfer PAMTRA

The Passive and Active Microwave TRAnsfer tool (PAMTRA) solves the radiative transfer for passive and active microwave radiation in all-sky conditions, i.e. cloudless, cloudy, and precipitating atmospheres (Mech et al., 2020). In this study, PAMTRA is used to simulate the brightness temperatures at the HATPRO frequencies during rain to investigate the impact of rain in the atmosphere and to assess the effect of liquid water accumulation on the radome (see Sec. 3.2).

## 3 Methodology

In this section, the problem of retrieving temperature profiles during rain is first shown using an example. Then the theoretical basics of how to create a temperature retrieval are explained. Finally, the procedure to select the most relevant frequencies and elevation angles is explained and the results of the information content analysis are shown. Figure 2 illustrates a time series of a HATPRO measurement in non-rainy and rainy conditions. The problem of state-of-the-art temperature retrievals during rain, indicated by unrealistic spikes is shown in Fig. 2 (d). The rain and sun quality flag (a) denotes if rain was detected by HATPRO's weather station or if the Sun or the Moon is directly in the receiver's field of view. Both would affect the quality of the retrieval. The second panel (b) shows the results of the spectral consistency check which is retrieved by the so-called *tbx* retrievals. Since the signal in the individual channels are highly dependent on each other, they can be used to retrieve the entire spectrum. During spectral consistency check (*tbx* retrievals), 13 of the 14 HATPRO frequencies are used to estimate the value of the unused frequency which is then compared to the measured brightness temperature and the discrepancy is noted. This procedure is repeated for all 14 frequencies. If the brightness temperature difference at a given frequency exceeds the limits of 1 K for K-band and 2 K for V-band, the time steps are flagged with *spectral consistency failed*. This is done here only for zenith observations and it usually happens when nonphysical or unrealistic spectra are measured due to rain or other obstacles in the field of view. During rainy periods none of the frequencies has passed the consistency check, therefore none of the frequencies are reliable to be used. Thus, the retrieval will not be trustworthy.

Figure 2 (c) shows the temperature variation and rainfall rate from the HATPRO weather station during the example day. There are no physical temperature gradients during rain events that might explain the height-time series of temperature (d). The shown temperature profiles are retrieved by the RPG firmware retrieval for Lindenberg which is based on a neural network approach using all 7 V-band frequencies and all 10 elevation angles. This frequency and elevation angle setup corresponds to the state of the art in determining temperature profiles under rain-free conditions.

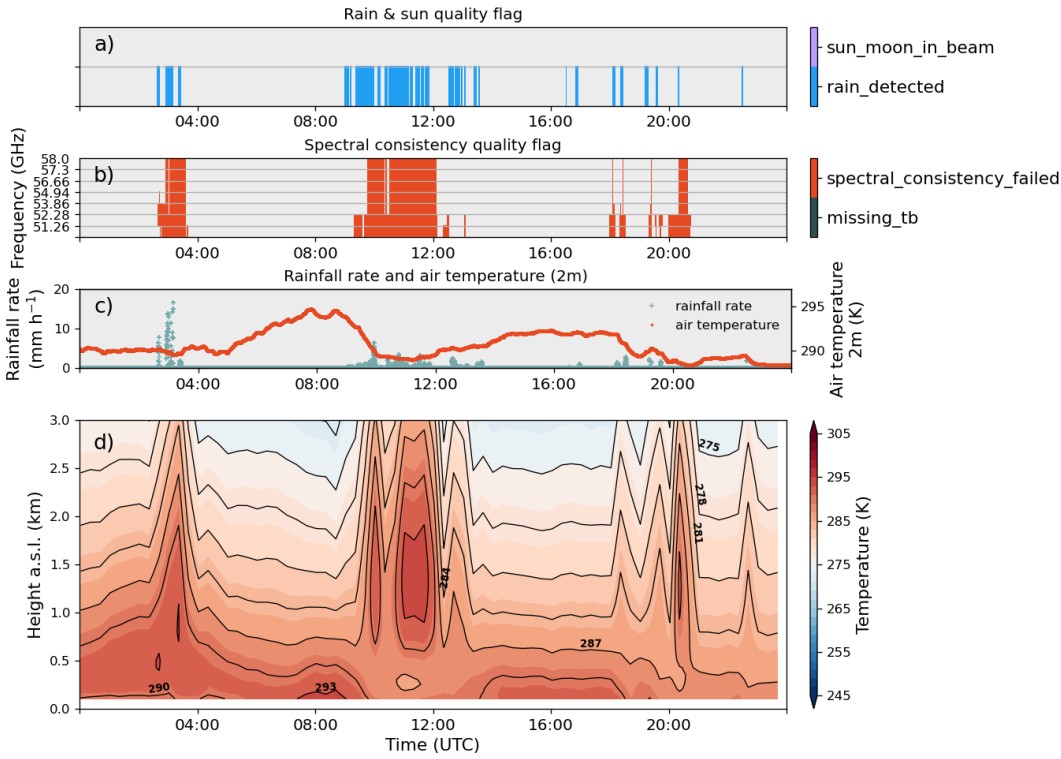

**Figure 2.** Time series of Moon or Sun and rain quality flag (a), spectral consistency quality flag (b), air temperature and rainfall rate from HATPRO's weather station (c), and height-time series of temperature profiles based on HATPRO's firmware radiometer retrieval algorithms in Lindenberg (Germany) on Aug 26, 2020. tb in the colorbar of panel (b) means brightness temperature.

All MWR retrievals, including *tbx* retrievals and temperature profile retrievals, need to be created for each specific geographic region, as typical atmospheric profiles of temperature and humidity vary across the globe. Walbröl et al. (2022) e.g. created MWR retrievals for low-humidity conditions in the Arctic and Schnitt et al. (2024) for the tropical Atlantic.

### 3.1 Temperature profile retrieval method

The retrieval essentially consists of a series of coefficients that can be based on an artificial neural network or a multi-linear regression model that relates modeled brightness temperatures and temperature profiles (Löhnert and Maier, 2012). In this work we use a regression model. The temperature profiles are based on 10 172 radiosondes and 173 088 ERA5 output profiles corresponding to the location of the MOL-RAO site in Lindenberg. We decided to use these two different data sources to get a data-set which contains profiles with high vertical resolution (radiosonde) and a large amount of profiles with modeled liquid water information (ERA5). From this data-set, temperature, humidity and pressure profiles are extracted. The cloud

**Table 1.** Retrieval specification. Zenith mode frequencies indicate the frequencies ($\nu$) that are observing only in zenith direction whereas scanning mode frequencies mark those measuring in the directions given by the elevation angle ($\varphi$) in the last column. Retrieval name nomenclature: $X\nu[z]Y\varphi$. $X$: number of frequencies with elevation scanning; $Y$: number of elevation angles. The index z indicates that, additionally, three zenith observations for 51.26-53.86 GHz have been included in retrieval development (first row). Nomenclature according to Crewell and Löhnert (2007).

|  | zenith mode frequencies (GHz) | scanning mode frequencies (GHz) | elevation angles (°) |
|---|---|---|---|
| $4\nu z10\varphi$ | 51.26, 52.28, 53.86 | 54.94, 56.66, 57.3, 58 | 90, 30, 19.2, 14.4, 11.4 8.4, 6.6, 5.4, 4.8, 4.2 |
| $4\nu 10\varphi$ | – | 54.94, 56.66, 57.3, 58 | 90, 30, 19.2, 14.4, 11.4 8.4, 6.6, 5.4, 4.8, 4.2 |
| $7\nu 9\varphi$ | – | 51.26, 52.28, 53.86 54.94, 56.66, 57.3, 58 | 30, 19.2, 14.4, 11.4 8.4, 6.6, 5.4, 4.8, 4.2 |
| $4\nu 9\varphi$ | – | 54.94, 56.66, 57.3, 58 | 30, 19.2, 14.4, 11.4 8.4, 6.6, 5.4, 4.8, 4.2 |

liquid water content is directly extracted from the ERA5 data. For the radiosonde data a cloud is synthetically determined where 95 % relative humidity is reached (Decker et al., 1978). The modified adiabatic liquid water content is then determined for the altitude range of the cloud according to Karstens et al. (1994). This information is used as input to the non-scattering

microwave radiative transfer model (see Sec.2.5). For each input profile the brightness temperatures which would be measured by a microwave radiometer under the given input conditions, frequencies and elevation angles are simulated. In total 146 608 profiles (80% randomly chosen profiles) were used for the training and 36 652 (20%) to test the regression model to predict the temperature profiles based on simulated brightness temperatures. In this study, different retrieval settings (varying number of frequencies and angles) were generated to contrast the RPG firmware method based on seven frequencies in the V-band

(oxygen complex) and ten elevation angles including the zenith direction (90°). Specifically, new retrieval setups are proposed that are only based on the upper four HATPRO frequencies in the V-band which exclude the zenith observation (nine angles). The different retrieval setups are listed in Tab. 1. The $4\nu z10\varphi$ retrieval is the most commonly used retrieval for low-level temperature profiling during non-rainy conditions. It uses 10 elevation angles (including the zenith angle) and the upper four frequencies of the V-band. Additionally, the lower three frequencies of the V-band are used at the zenith angle.

The question how the frequencies and elevation angles for the new temperature retrievals are selected is discussed in the following subsection. The performance during non-raining (cloudy and cloudless) conditions is treated in Sec. 4.1 and is illustrated in Fig. 6.

## 3.2 Selection of frequencies and elevation angles

To select frequencies and elevations angles for a new temperature retrieval that is less compromised by rain, it is necessary to check which frequencies are less affected by rain accumulated on the radome and by rain in the atmosphere. This was done by a special MWR measurement strategy during a rain event described below. It is worth noting again, that during rain, the atmosphere becomes more opaque with increasing frequency in the V-band.

On July 27, 2023, and on August 1, 2023, on the roof measurement platform of the Institute for Meteorology of Leipzig University, special measurements were performed with the microwave radiometer HATPRO. There was continuous rain from 9:00 to 15:00 UTC with rain rates, observed by HATPRO weather station, generally below $2\,\mathrm{mm\,h^{-1}}$ on July 27 followed by showers with low intensities. On August 1, it rained continuously from midnight to 8:30 UTC with rain rates generally below $2\,\mathrm{mm\,h^{-1}}$ but occasionally reaching $7\,\mathrm{mm\,h^{-1}}$. Afterwards, there were repeated rain showers and cloudless periods until the end of the day. On July 27 at 7:01 UTC as well as on August 1 at 7:41 UTC and 14:14 UTC scan patterns from $0°$ (horizontal) to $90°$ (zenith) with $5°$ elevation angle steps were carried out. In addition, PAMTRA simulations of brightness temperatures at all specified elevation angles were carried out for the three different situations on these days: no rain with a thin ice cloud (July 27), moderate rain ($5.5\,\mathrm{mm\,h^{-1}}$, Aug 1) and very heavy rain ($61\,\mathrm{mm\,h^{-1}}$, Aug 1). The ECMWF model output profiles of temperature, pressure and relative humidity in Leipzig from the same day were taken as input for the simulations. Rain drop size distributions for the stratiform rain event early in the day and for the heavy rain shower around 14:14 UTC were estimated by a modified gamma distribution ($\mu = 2$, $\gamma = 1$) with rain water contents of $0.23\,\mathrm{g\,kg^{-1}}$ and $1.6\,\mathrm{g\,kg^{-1}}$ and number concentrations of $400\,\mathrm{m^{-3}}$ and $30\,\mathrm{m^{-3}}$, respectively, with uniform rain drop size distributions between cloud base of 2.5 km and the surface. The rain drop size distribution were chosen in a way such that the simulated rain rates match the observations.

The brightness temperatures from HATPRO observations and from PAMTRA simulations (a,b,c) as well as their difference (d,e,f) as a function of the elevation angle are illustrated in Fig. 3 for the seven frequencies in the V-band and for three weather conditions (no rain, moderate rain, very heavy rain). It can be seen that the simulation and observation fit well for the profile with no rain at 7:01 UTC on July 27. The differences of around 6 K on average for the lower frequencies and higher elevation angles might be caused by the ECMWF model input which slightly differs from the atmospheric state that was observed by MWR. Additionally, for atmospheric boundary layer scan homogeneous conditions are assumed. If this is not the case, different air masses might be observed by the more transparent channels at 51.26, 52.28, and 53.86 GHz. For the profile at 7:41 UTC on August 1 with rain rates of $5.5\,\mathrm{mm\,h^{-1}}$ (observed) and $5.3\,\mathrm{mm\,h^{-1}}$ (simulated) the brightness temperatures from 51.26, 52.28, and 53.86 GHz differ from the simulation above $45°$ elevation angle by up to 26, 18, and 6 K, respectively. This might be caused by the accumulation of liquid water from rain on the top of the MWR radome. For the heavy rain shower at 14:14 UTC on August 1 with rain rates of $61.1\,\mathrm{mm\,h^{-1}}$ (observed) and $61.7\,\mathrm{mm\,h^{-1}}$ (simulated) the simulated and the observed brightness temperatures at the same three frequencies differ by up to 36, 28 and 10 K, respectively, above $40°$ elevation angle. 54.94, 56.66, 57.3, and 58 GHz as well as all angles below $45°$ are apparently unaffected by the impact of rain and show no significant difference between simulated and observed brightness temperatures. The range of brightness temperature difference at the lower elevation angles (below $45°$) is roughly around -5 to 5 K. When the brightness temperature

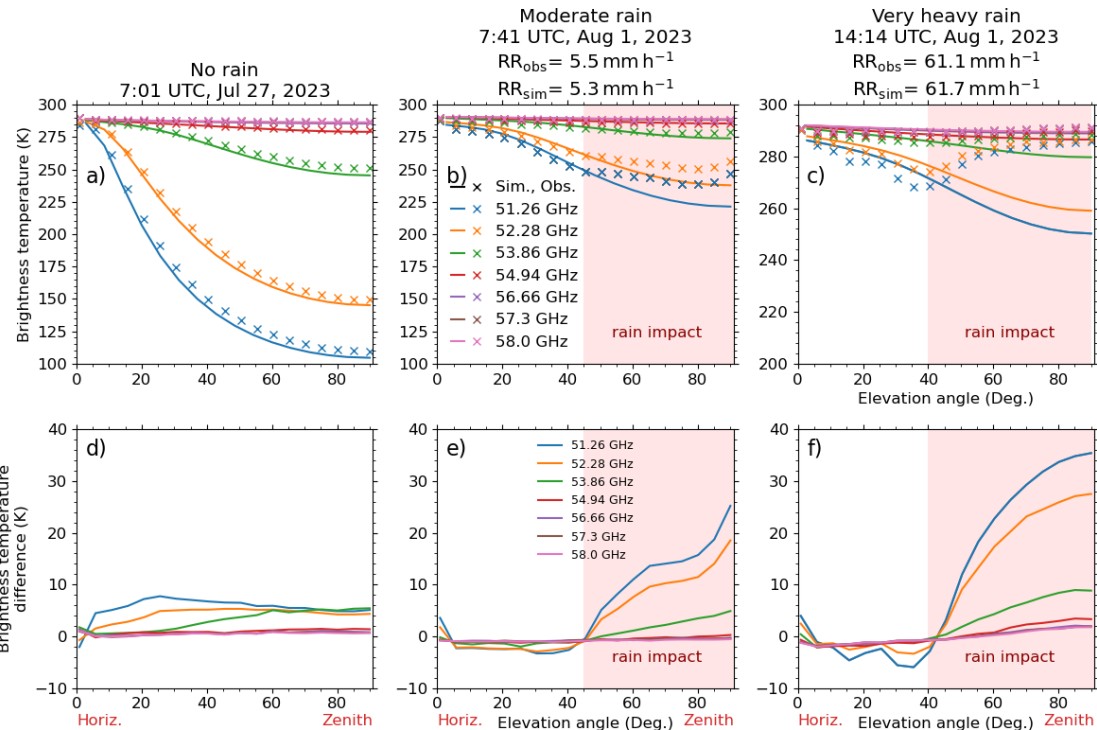

**Figure 3.** Observed and simulated brightness temperatures (a,b,c) as well as their difference (d,e,f) for different frequencies (colors) versus elevation angle for no rain (a, d), moderate rain (b,e), and very heavy rain events (c,f). Rose rectangle marks the area where the observations significantly differ from the simulation probably caused by wet radome. Note that the y-axis in (c) differs from (a) and (b).

difference exceeds this range, this is defined here as significant deviation. That means that all elevation angles below 40° and the upper four HATPRO frequencies from the V-band can be used to retrieve temperature profiles during rain. It is important to note that most state-of-the-art temperature retrievals from atmospheric boundary layer scans (e.g. HATPRO's firmware) uses the set of elevation angles shown in Fig. 1, thus the majority of elevation angles used by the retrievals are below 40° except for the zenith observation.

The spectral consistency check applied to all elevation angles using the corresponding *tbx* retrievals for these angles shows similar results. Figure 4 illustrates the 95th quantile of the brightness temperature difference (observed – retrieved) for different elevation angles for all elevation scans that were performed during rain in the observation period. The 95th quantile is used here to exclude outliers and has more significance than median or mean. For 95% of the zenith observations the difference is larger than 2 K for all frequencies except for 58 GHz and even for small rain rates. The 58 GHz channel at zenith observation (a) shows small differences since this channel is almost saturated which means that even rain does not increase the observed signal significantly. A typical threshold used for the maximum allowed difference would be 2 K as used in the open source processing software MWRpy (Marke et al., 2024). Values higher than 2 K indicate inconsistency in the spectrum probably

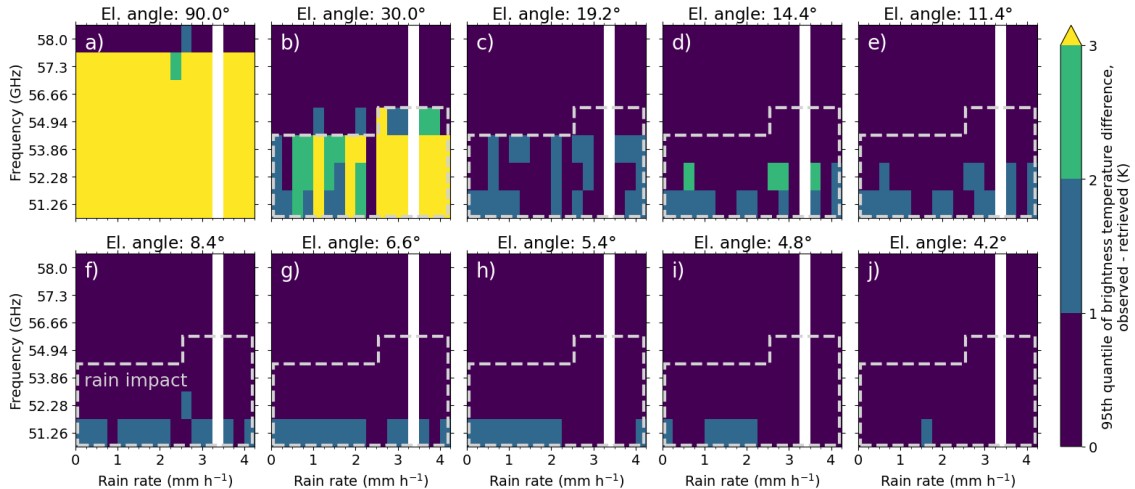

**Figure 4.** 95th quantile of brightness temperature difference (observed – retrieved) per frequency (y-axis) and rain rate (x-axis) for different elevation angles (a-j). The grey dashed boxes mark the area of rain impact determined by differences of more than 2 K.

caused by rain. For the $30\,^\circ$ elevation angle the differences are larger than 3 K for rain rates above $2.5\,\mathrm{mm\,h^{-1}}$ and for the first three frequencies of the V-band. Lower elevation angles (below $19.2\,^\circ$) show smaller differences in the brightness temperature and mostly below 2 K for all rain rates and frequencies, except the 52.28 GHz channel at $14.4\,^\circ$. This implies that the upper four frequencies of the V-band can be used for temperature retrievals at elevation angles below $30\,^\circ$ for rain rates up to $2.5\,\mathrm{mm\,h^{-1}}$. Disturbances of the observaions by a wet radome would result in larger differences as can be seen at the zenith angle ($90\,^\circ$).

One might expect that adding the lower HATPRO frequencies of the V-band (i.e. using all seven frequencies in the retrieval) would be more appropriate, as the atmosphere is more transparent at these frequencies. However, our analyses have shown that for that purpose the lower V-band frequencies are not optimal and instead increase uncertainties. Horizontally homogeneous conditions are assumed for boundary layer scans. At low elevation angles, however, different air masses are observed by the more transparent channels leading to uncertainties in the retrieved profiles.

## 3.3 Information content analysis

To investigate how much information originates from the observations and not from the climatology, an optimal estimation technique has been applied to the case studies (Rodgers, 2000; Maahn et al., 2020). It calculates the degrees of freedom of a signal (DFS) and specifies the information content that comes from the measurement itself. The cumulated DFS of all four retrievals are illustrated in Fig. 5 for the three weather conditions (a, b, c) mentioned in Sec. 3.2. The curves of all retrievals have a similar shape and do not differ significantly below heights of 3 km. However, with increasing altitude, the differences of cumulative DFS increase. Once a DFS curve reaches a vertical line no more information is added by the measurements. The retrievals with fewer frequencies and angles ($4\nu10\varphi$, $4\nu9\varphi$) display lower values of the cumulated DFS under all three weather conditions. This means that there is less information from altitudes above roughly 1.5 km from the measurement and

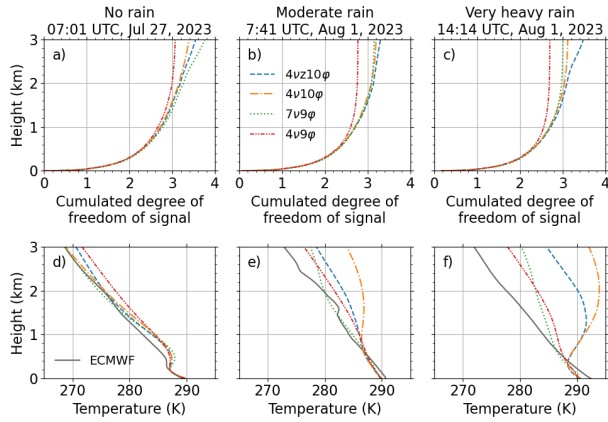

**Figure 5.** The cumulated degrees of freedom of signal and temperature profiles for no rain (a, d) conditions on Jul 27, 2023, moderate rain (b, e) and heavy rain (c, f) conditions on Aug 1, 2023.

the profile is more driven by the climatology. The more rain there is in the atmosphere, the lower the information content of the measurement, as can be seen in the maximum value of the cumulated DFS in 3 km which reaches values between 3 and 4 for no rain and between 2.8 and 3.4 during moderate rain and between 2.7 and 3.4 during very heavy rain. Summarizing, the retrieved temperature profile is driven by the measurement at least up to 3 km for no rain, about 1.5 km for rain and about 1 km for heavy rain proven by the determined DFS indicated by the point at which the line with lowest information content (red) becomes vertical.

The retrieved temperature profiles from the four retrievals, as well as the ECMWF temperature output profile for the same three conditions (no rain, moderate rain, very heavy rain) are illustrated in Fig. 5(d, e, f). As expected for non-rainy conditions (d) and shown in section 4.1, all four retrievals show similar deviations from the reference ECMWF profile in the lowest 1.5 km. Above 1.5 km the $4\nu9\varphi$ differs from $4\nu z10\varphi$, $4\nu10\varphi$ and $4\nu9\varphi$ as well as from ECMWF output. For the moderate rain case (e), all retrievals perform similar below about 1 km. Retrievals which use zenith observations ($4\nu z10\varphi$ and $4\nu10\varphi$) perform worse than the others ($7\nu9\varphi$ and $4\nu9\varphi$). The $7\nu9\varphi$ retrieval performs best and shows smallest differences to the ECMWF profile with a difference of 1 K below 2 km. For the very heavy rain event (f), the $7\nu9\varphi$ and $4\nu9\varphi$ retrievals shows the best performance indicated by the smallest difference to the reference ECMWF model output. As expected $4\nu z10\varphi$ and $4\nu10\varphi$ have largest deviations (more than 12 K in 2 km) from ECMWF model output since they are intentionally made for non-rainy conditions. It is likely that the ECMWF temperature profile does not represent the truth, especially during rain showers. For this reason, no quantitative statement is made here and more attention is paid to the intercomparison between the individual retrievals.

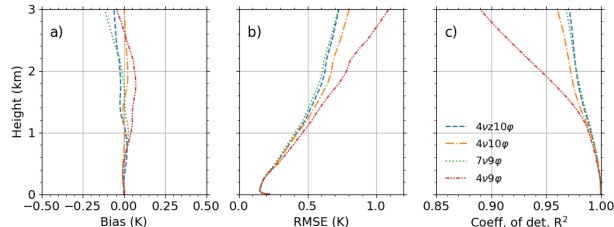

**Figure 6.** Temperature retrieval performance in terms of bias (a), root mean square error (RMSE, b) and coefficient of determination (c) based on synthetic data (trained with radio-soundings and ERA5) during cloudy and cloudless conditions.

## 4   Results

This section first shows the performance of the newly created temperature profile retrievals based on simulations with the test data-set under non-rainy conditions. This is only to show that the new different retrievals produce meaningful results. In section 4.2, the retrieval performance is evaluated on the basis of observations using the MOL-RAO case study of Aug 26, 2020, introduced in Sec. 3. Finally, the retrieved temperature profiles are compared to ECMWF output on a larger data-set.

### 4.1   Retrieval performance based on simulations during non-raining conditions

The performance of the new approaches ($7\nu10\varphi$, $4\nu10\varphi$, $4\nu9\varphi$) in comparison to the common retrieval ($4\nu z10\varphi$) under non-raining idealized conditions is shown in Fig. 6. This is the result of the test data from the atmospheric profiles from radiosonde and ERA5 (36 552 profiles). Bias (a), root mean square error (RMSE, b), and coefficient of determination ($R^2$, c) between true values and the prediction of the regression model indicate how much uncertainty is added by omitting frequencies and elevation angles during cloudy and cloudless conditions using profiles from the test data-set. All four sets of retrievals show similar behavior in bias (a), namely just small systematic deviations from zero. For all four retrievals, RMSE (b) increases with altitude while $R^2$ decreases with altitude, both indicating an increase in uncertainty with height. RMSE and $R^2$ diverge above 1 km with $4\nu9\varphi$ being worse whereas $4\nu z10\varphi$, $7\nu10\varphi$ and $4\nu10\varphi$ almost overlap. Bias, RMSE and $R^2$ values are in accordance with Crewell and Löhnert (2007). Highest uncertainties are evident for the $4\nu9\varphi$ retrieval. This is an expected behavior since information can be lost by omitting frequencies and zenith observations as shown in Fig. 5. In conclusion, the $4\nu9\varphi$ retrieval does not perform as well as the other retrievals which is expected as it was optimized for rainy conditions.

### 4.2   Case study based on observations

The four temperature profile retrievals introduced in Tab. 1 were applied to the MOL-RAO example of Aug 26, 2020. Results are displayed in Figure 7 where the height-time plots of the ECMWF model temperature (a), the four temperature retrievals (b, d, f, h), as well as the difference of the retrieved temperatures to the ECMWF model temperature (c, e, g, i) are shown. As introduced above (Sec. 2.1) there are three rain events on that day, early morning around 03 UTC, between 09 and 13 UTC

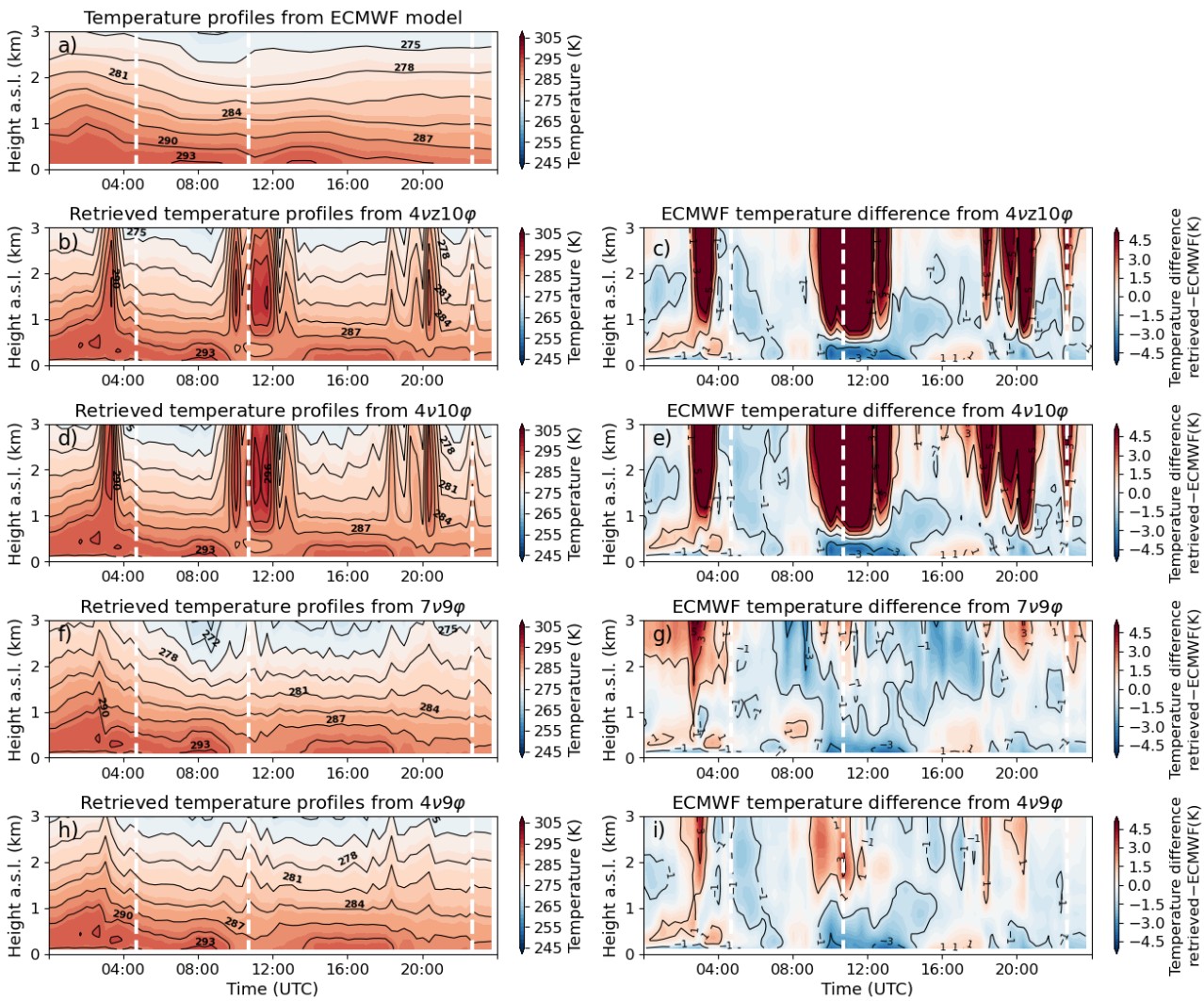

**Figure 7.** Height-time series of temperature profiles from ECMWF model (a) and temperature profiles based on different retrieval algorithms (b, d, f, h) and associated temperature difference to ECMWF model (c, e, g, i) in Lindenberg (Germany) on Aug 26, 2020. The radiosonde launch times are indicated by white dashed lines.

and around 20 UTC (see Fig. 2 a). During all rain events with rain rates between 0 and at maximum $10\,\mathrm{mm\,h^{-1}}$ the spectral consistency check failed (Fig. 2 b). The presence of rain in the lower atmosphere or accumulated liquid water on the radome

compromises the retrieval output indicated by the unrealistic spikes in the temperature profiles (Fig. 7 b, d) and by a high temperature difference (c, e). Neither the $4\nu z10\varphi$ nor the $4\nu 10\varphi$ can be applied during rain conditions, as can be seen by very large positive temperature differences of more then $10\,\mathrm{K}$ above $1\,\mathrm{km}$ and values below $-3\,\mathrm{K}$ below $1\,\mathrm{km}$ during a the rain events. However, the $7\nu 9\varphi$ as well as the $4\nu 9\varphi$ retrieval can tackle the rain limitation and are able to produce reasonable results in comparison to the ECMWF model temperature output (f, g, h, i). Their deviations are mostly below $3\,\mathrm{K}$ during rain

and mostly between $-1$ and $1\,\mathrm{K}$ for the rest of the day. Nevertheless, during the rain events there is some variability in the $7\nu 9\varphi$ and $4\nu 9\varphi$ retrievals in contrast to the ECMWF profile. This is probably caused by a wet radome as the rain rates are larger than $2.5\,\mathrm{mm\,h^{-1}}$ (see Fig. 2) which is the threshold derived in Fig. 4.

To further estimate the performance of the four temperature profile retrievals, they are compared to radiosonde launches at MOL-RAO on Aug 26, 2020, 04:45 (a), 10:45 (b) and 22:25 UTC (c) in Figure 8. During the launch at 04:45 UTC in

non-raining conditions there are no significant differences between the retrievals, the sounding and the ECMWF temperature profiles (a). The differences are much higher during the rain event at 10:45 UTC (b) with rain rates around $1.5\,\mathrm{mm\,h^{-1}}$. Sounding and ECMWF model temperature profile are in good agreement and only the $7\nu 9\varphi$ and the $4\nu 9\varphi$ retrievals fit the sounding as reference within less than $2\,\mathrm{K}$ below $1\,\mathrm{km}$. Above $1\,\mathrm{km}$ the $7\nu 9\varphi$ retrieval performs best, since it almost overlaps with the sounding. The $4\nu 9\varphi$ retrieval deviates around $3\,\mathrm{K}$ at $2\,\mathrm{km}$. In contrast, the temperature retrievals from $4\nu z10\varphi$ and

$7\nu 10\varphi$, are completely off by over $10\,\mathrm{K}$ above $1\,\mathrm{km}$. The temperature profile comparison during the short and light rain shower with rain rates below $0.5\,\mathrm{mm\,h^{-1}}$ at around 22:45 UTC in Fig. 8 (c) shows a similar result, the $7\nu 9\varphi$ and the $4\nu 9\varphi$ retrieval even fit to the reference sounding within the expected sounding uncertainty.

Up to this point, the performance of the retrieval has been evaluated only on the basis of case studies. In the next section it will be evaluated against ECMWF model output using a larger data set.

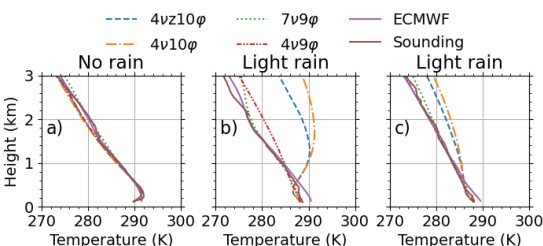

**Figure 8.** Panels a, b, and c show a comparison of three retrieved temperature profiles obtained from the four different retrievals with radiosoundings launched at MOL-RAO on Aug 26, 2020, at 4:45 (a), 10:45 (b), and 22:45 UTC (c) and ECMWF model output from 5, 11, and 23 UTC.

## 4.3 ECMWF model comparison

In this section the performance of the $4\nu9\varphi$, $7\nu9\varphi$ and the state-of-the-art $4\nu z10\varphi$ temperature retrieval against ECMWF model temperature profiles is investigated. Therefore, all three months of HATPRO observation at MOL-RAO from July to October 2020 are taken into account. Hourly ECMWF model temperatures are interpolated to the measurement grid of approximately 20 minutes per temperature profile, since there is a routine elevation scan every 20 minutes. Figure 9 shows the retrieval performance in terms of bias (left panels) and root mean square errors (RMSE, right panels) between ECMWF output and retrievals for non-raining cases (a, b), and raining cases with rain rates smaller than $0.5\,\mathrm{mm\,h^{-1}}$ (c, d), rain rates between 0.5 and $2.5\,\mathrm{mm\,h^{-1}}$ (e, f) and rain rates larger that $2.5\,\mathrm{mm\,h^{-1}}$ (g, h). The rain rates used here, are from the HATPRO weather station. During non-raining conditions (3671 sample profiles) all retrievals agree well with the ECMWF output (Fig. 9 a, b, c, dashed) as could be expected from Fig. 6. But for small rain rates below $0.5\,\mathrm{mm\,h^{-1}}$ (57 sample profiles) the proposed $4\nu9\varphi$ agrees much better with a bias of around 1 K (Fig. 9 c, dash-dot) and a RMSE ranging between 0.5 and 2 K (d). The state-of-the-art retrieval ($4\nu z10\varphi$) leads to very high deviations from the ECMWF temperature profiles with biases and RMSE's around 5 to 7 K and 5 to 10 K, respectively, apart from altitudes below 0.5 km. The bias of the $7\nu9\varphi$ and the $4\nu9\varphi$ retrievals increase with height and reach a maximum values of around 4 K in 3 km for rain rates between 0.5 and $2.5\,\mathrm{mm\,h^{-1}}$. The corresponding RMSE's are around 1.5 K within the lowest 1 km and increase up to around 5 K at 3 km. For rain rates above $2.5\,\mathrm{mm\,h^{-1}}$ the biases and RMSE are largest for each retrieval. The higher the rain rate, the worse the performance of the MWR temperature profile retrievals. Although the $7\nu9\varphi$ and $4\nu9\varphi$ are significantly better than the common $4\nu z10\varphi$ retrieval, they deviate from the ECMWF output. Of course the ECMWF model output is not the truth, especially during rain, but serves as a reference for comparing the three retrievals. It should be noted that the $7\nu9\varphi$ and $4\nu9\varphi$ retrievals perform better since the common $4\nu z10\varphi$ retrieval setup was intentionally not developed for working under raining conditions. Summarizing, that the new proposed retrieval based on MWR observation under lower elevation angles and only the higher V-band frequencies allows to resolve temperature profiles during rain with rain rates up to $2.5\,\mathrm{mm\,h^{-1}}$ which was not possible before with the state-of-the-art retrievals.

## 5 Conclusions and Outlook

In summary, the HATPRO $4\nu9\varphi$ retrieval method demonstrated in this study achieves unprecedented accuracy of low-level temperature profiling with a bias of less than 1.5 K and an RMSE below 2 K up to 3 km in rain with rain rates below $0.5\,\mathrm{mm\,h^{-1}}$ compared to ECMWF temperature profiles. For rain rates between 0.5 and $2.5\,\mathrm{mm\,h^{-1}}$ the bias increases up to 2 K and RMSE up to 3 K in 1.5 km. An intercomparison of the different retrievals during non-raining conditions showed a good agreement in bias and RMSE values, respectively. As shown based on ERA5 and radiosonde data, the proposed $4\nu9\varphi$ retrieval performs very similar to the state-of-the-art $4\nu z10\varphi$ retrieval up to 1.5 km, during non-rainy conditions. Above these heights, the RMSE increases up to 1.2 K instead of 0.8 K in 3 km as the $4\nu z10\varphi$, $7\nu10\varphi$ and $4\nu10\varphi$ retrievals which almost overlap. The bias is very similar to the stat-of-the-art retrieval around zero from surface up to 3 km. It was shown that even in very heavy rain ($61\,\mathrm{mm\,h^{-1}}$) measurements at elevation angles below $40°$ can be used to derive temperature profiles up to 1.5 km using the

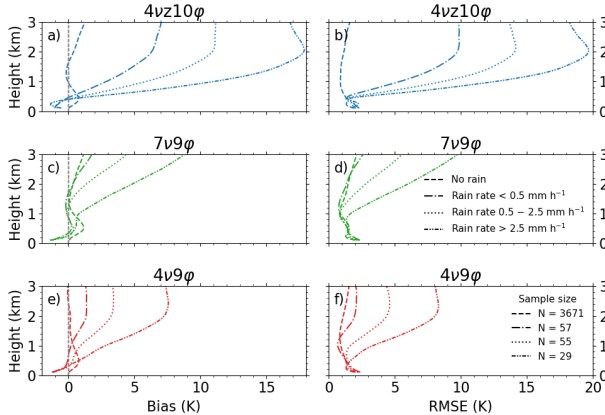

**Figure 9.** Bias (left panel) and root mean square error (right panel) between retrieved and ECMWF temperature profiles for rain free and rain cases with different rain rates (lines) and for different retrievals (rows, colors). N denotes the number of time steps taken into account at German Weather Service Observatory in Lindenberg (MOL-RAO) between 16 Jul, 2020 and 8 Oct, 2020. Bias is defined as retrieved minus ECMWF output as reference.

$4\nu9\varphi$. The $7\nu9\varphi$ partially performs better than the suggested $4\nu9\varphi$, but in general the $4\nu9\varphi$ is proposed to be used in most cases. The lower frequencies of the V-band used in the $7\nu10\varphi$ are more transparent and hence observe different air masses in the lower elevation angles which might lead to large uncertainties especially in the case of spatially variable precipitation. The recommendation is to use the $4\nu9\varphi$ retrieval for rain rates below $2.5\,\mathrm{mm\,h^{-1}}$ to retrieve temperature profiles up to 1.5 km with uncertainties less than 2 K.

The temperature retrievals can be easily applied with an existing open source software (MWRpy). In addition, the published software package can be used to create custom retrievals for user-defined locations (Foth, 2024b). This represents a significant improvement towards the reliability of using MWR for weather nowcasting or forecast. Improved low-level temperature profile retrievals are of great values for the following applications: investigations of evaporative cooling during precipitation evaporation can be improved by more accurate temperature profile retrievals which can in turn improve the reliability of the evaluation of model parameterizations. Furthermore, the proposed method can be applied retrospectively to correct temperature profiles from long-term observations as long as the MWR scanning brightness temperature data is available for the post-processing. In this way improved climatologies of MWR-based temperature profiles can be derived.

Several future modifications to even increase the performance of the presented retrieval are envisioned: an optimal estimation method which is also a variational technique could be used in further investigations. In contrast to Cimini et al. (2011), only HATPRO frequencies that pass the consistency check for all elevation angles should be used at each time step independent of the rain situation. Thus, a continuous time series of temperature profiles can be created, which provides physical uncertainties for each time and height range. This might also improve profiles of absolute humidity which is also of interest for evaporation

studies. Additionally, long-term HATPRO observations will enable a quantification of the maximum rain rate at which the new $4\nu9\varphi$ retrieval can be applied.

*Code and data availability.* The HATPRO raw data is processed with MWRpy version 0.8.2 (https://github.com/actris-cloudnet/mwrpy). Also some MWRpy subroutines for plotting are used in this study. The optimal estimation software package pyOptimalEstimation version

1.2 is available under https://github.com/maahn/pyOptimalEstimation and described in detail in Maahn et al. (2020). The Passive and Active Microwave TRAnsfer model PAMTRA is also available on github.com (https://github.com/igmk/pamtra) and is already published in Mech et al. (2019, 2020). ERA5 data is available under https://cds.climate.copernicus.eu/ (Hersbach et al., 2019). The HATPRO data from the general scans in Leipzig is available at zenodo (Foth, 2024a). The Lindenberg HATPRO and model data used in this study are generated by the Aerosol, Clouds and Trace Gases Research Infrastructure (ACTRIS) and are available from the ACTRIS Data Centre using the

following links: https://doi.org/10.60656/ca8017ee6ef94027, https://doi.org/10.60656/E938967BC0524DEE. The retrievals are made with the pyMakeRetrieval routines version 1.2.0 (Foth, 2024b) and are available on github (https://github.com/remsens-lim/pyMakeRetrieval).

*Author contributions.* AF prepared the manuscript in close cooperation with ML, PSG and HKL. AF performed the investigations and data analyses. ML and AF realized the experimental setup in Lindenberg and Leipzig, respectively, and were responsible for the high quality of the HATPRO measurements. The conceptualization was initialized by AF, ML and PSG. All authors have contributed to the scientific

discussions.

*Competing interests.* The authors declare that they have no conflict of interest.

*Acknowledgements.* The authors thank the LIM-team and the MOL-RAO team for supporting the HATPRO observations in Lindenberg. The authors also acknowledge the ACTRIS-Cloudnet team and all associated developers for the well documented code around remote sensing especially the HATPRO processing within MWRpy. This research has been supported by the Deutsche Forschungsgemeinschaft (DFG,

German Research Foundation) – project no. 268020496 – TRR 172, within the Transregional Collaborative Research Center "ArctiC Amplification: Climate Relevant Atmospheric and SurfaCe Processes, and Feedback Mechanisms (AC)3", sub-project B07 (grant 437153667) and E05, and grant number FO 1285/2-11. This research has been supported by the Federal State of Saxony and the European Social Fund (ESF) in the framework of the program "Projects in the fields of higher education and research" (grant no. 232101734 and 100339509). This work was funded by the Saxon State Ministry for Science, Culture and Tourism (SMWK) – [3-7304/44/4-2023/8846].

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
