# Peer review of "Determination of low-level temperature profiles from microwave radiometer observations during rain"

_EGUsphere, 2024_

## Referee Comment (RC1)

Review for „**Determination of low-level temperature profiles from microwave radiometer observations during rain**" submitted to *Atmospheric Measurement Techniques* by Foth et al.

**General comments:**

The manuscript analyses the performance of temperature profile observations by microwave radiometers during rain. This is a relevant topic as more and more of these instruments are operated continuously, and rainy profiles were usually discarded in the past.

I would, however, like to see a more general recommendation on how and under which circumstances these observations can be used. In the discussion, it needs to be clearly stated which channel/angle combination should be used during operation, and which uncertainties are considered to be acceptable, depending on height above ground and rain rate.

Concerning the radiative transfer calculations, I would strongly recommend using the updated Rosenkranz gas absorption model from 2022/2023, as this version significantly improves the radiative transfer results in the lower V-band.

**Specific comments:**

Lines 29ff: Please provide a more precise statement about which "high frequencies" become opaque? The center V-band frequencies (around 58 GHz) are already opaque without rain.

Lines 46ff: This motivation can be formulated better, such as: "The method presented here can be applied to standard measurement modes and does not require any changes in measurement setup".

Lines 63ff: HATPRO doesn't measure voltages, the detectors convert the antenna signal into voltages. A calibration is necessary to convert the voltages into brightness temperatures (but not "the voltages are then calibrated to brightness temperatures")

Line 69: Add: "during rain"

Section 2.2.: Why did you base your retrieval training on old RS80 radiosondes? These sondes are known to have a dry bias, and therefore might also bias your retrieval. I would strongly recommend using state-of-the-art sondes (Vaisala RS41) which have been used in Lindenberg for quite some time already.

Section 2.4: It seems you did not use all hourly data from 2000 to 2019 for the retrieval development. That would roughly make 20*365*24=175200 time steps, while you used 58195 profiles. How did you select the data? Are the data evenly distributed over the years and seasons?

Section 3, lines 117ff: Please explain better how spectral retrievals work: Individual channels are highly dependent on each other and can thus be used to retrieve the whole spectrum. Did you apply the spectral consistency check only for zenith observations or for all elevation angles? Please comment on that!

Fig. 3: The y-axis range until +/-60 K makes it difficult to see differences between the channels, especially around 0. I would recommend limiting the range to +/-20 K. By the way, I don't think that 10 K bias for some channels in the no-rain case (3a) is satisfactory, whereas in 3b for low

elevation angles, the agreement is much better. Do you have an explanation for this? For a bias in the input model data, I would expect that the difference changes with elevation angle.

Fig.3, continued: Furthermore, I couldn't find a clear explanation of the results in 3c, concerning the different fractions of rain used. Can you discuss that a bit more? Don't you consider the solid lines for 51.26 and 52.28 GHz (yellow, blue) for low elevation angles as significant deviations? Concerning the figure style: I cannot really distinguish the different lines in 3c neither.

Page 8, lines 191ff: This paragraph is a bit confusing, try to make your statements more clearly.

Figure 4: Please provide a somewhat larger plot! (same for Fig. 7)

Section 4.3: Did you classify the rainy cases using rain rates from the model or from observations?

Figure 8: I would strongly recommend to add a comparative figure (putting the red lines from a,c,e,g and b,d,f,h respectively into one figure). Like this, the additional uncertainties depending on the rain rate can be seen much more clearly. What about the performance of *7v9ϕ* ? I would be interested to see this combination in comparison here as well.

---

## Referee Comment (RC2)

I have reviewed the paper titled "Determination of low-level temperature profiles from microwave radiometer observations during rain" by Foth, Lochmann, Saaverdra Garfias and Kalesse-Los. I found the science and its presentation to meet the standards. I hence recommend that this manuscript be accepted with corrections.

Main questions/ concerns:

1.      L.65-69: discussion on the wet-radome mitigations on the HATPRO radiometer:

What is the HATPRO radome made of? Is it still the blue foam? if so, if it gets wet, it will take time to dry out and, as with sponges, the water will not stop at the top and will fill all the foam by capillarity. This could be problematic for measurements below zenith…

[Figure]

2. L.166, L.278 and L.281: On L. 166: "[…] no rain, moderate rain (2.7 mm h$^{-1}$ ) and heavy rain (3.7 – 11 mm h$^{-1}$ )". Is moderate defined as 2.7 to 3.6mm/h so that heavy will be defined as 3.7 to 11 mm/h and therefore light will be <2.6 mm/h? your definition is not clear to me. In North America, we have definition for light, moderate and heavy rains: light (< 2.5 mm/h), moderate (2.6 to 7.5 mm/h) and heavy (> 7.6 mm/h); there might be relevant standards in Europe to follow.

Although I agree that the new 4ν9φ scanning strategy can be used up to 2mm/h (L.278) without too much bias, the following lines (L. 280 – 282) in the conclusion:

"In summary, the HATPRO 4ν9φ retrieval method demonstrated in this study achieves unprecedented accuracy of low-level temperature profiling up to 2 km in rain. It was shown that even in heavier rain measurements at elevation angles below 40$^{\circ}$ can be used to derive temperature profiles up to 1.5 km. "

could be interpreted very differently should a casual reader only read the concluding remarks… As such, I would strongly advise that the sentence be re-written much clearer with the

limitations of 2mm/h. It is also good practice, even for short papers, to summarize the findings in the conclusion for the casual reader.

3. L.250 & Figure 6 : Although I agree that the 4ν9φ outperforms the other retrievals, there is still some clear influence of the rain in the temperature measurements as seen in Figure 6.h: the retrieved temperature between rain events is smooth like the ECMWF, but during the rain events (3 UTC, 9-13 UTC and 20UTC) the retrieved temperature still shows quite some variability compared to the temperature profiles and still leads to +/- 3K temperature difference. This could be because of the wet radome, as the wet radome emissions are likely angle dependent.

Minor fixes:

-       L. 35 – 37: "Xu et al. (2014) retrieved thermodynamic profiles such as temperature and humidity as well as liquid water profiles by using off-zenith MWR observations at $15^{\circ}$ elevation to reduce the impact of rain on the measurements. As retrieval technique Xu et al. (2014) used a neural network approach. "

would be clearer if written:
Xu et al. (2014) retrieved thermodynamic profiles such as temperature and humidity as well as liquid water profiles by using off-zenith MWR observations at $15^{\circ}$ elevation to reduce the impact of rain on the measurements [using] a neural network approach.

-       L. 82: "Their accuracy in contrast to other types of radiosondes is described by Turner et al. (2003). "

It is unclear to me where this phrase is going. How does the accuracy of the RS80 compare to other types namely the RS41 which is used by the Lindenberg site (MOL-RAO).

-       L. 91-95: there is not a single reference to ERA5 papers.  At the very least the following paper should be referenced:

Hersbach, H. et al.  2020: The ERA5 global reanalysis. Q. J. of the R. Meteorol. Soc., 146, 1999–2049.https://doi.org/10.1002/qj.3803.

-       L. 118 – 120: "The second panel (b) shows the results of the spectral consistency checks which is retrieved by the so-called *tbx* retrievals, which work as follows. There are 14 HATPRO frequencies and only 13 of them are used to estimate the expected value for the 14th

frequency and then the difference between the estimated and the measured brightness temperature is determined. "

would be clearer if written:

The second panel (b) shows the results of the spectral consistency check which is retrieved by the so-called *tbx* retrieval [. During spectral consistency check (*tbx* retrievals), 13 of the 14 HATPRO frequencies are used to estimate the value of the unused frequency which is then compared to the measured brightness temperature and the discrepancy is noted.]

-       Figure caption on Figure 2. "Time series of Moon or Sun and rain quality flag (a), spectral consistency quality flag (b), air temperature and rainfall rate from HATPRO's weather station (c), and height-time series of temperature profiles based HATPRO's firmware radiometer retrieval algorithms in Lindenberg (Germany) on Aug 26, 2020. tb in the colorbar (b) means brightness temperature. "

Should be:

Time series of Moon or Sun and rain quality flag (a), spectral consistency quality flag (b), air temperature and rainfall rate from HATPRO's weather station (c), and height-time series of temperature profiles based HATPRO's firmware radiometer retrieval algorithms in Lindenberg (Germany) on Aug 26, 2020. tb in the colorbar ( [d] ) means brightness temperature.

---

## Author Comment (AC1)

**Response to Reviewers #1, #2 and #3**

We like to thank the reviewers for providing helpful comments to improve the manuscript.

We made substantial improvements according to your suggestions. All changes are highlighted in the diff-manuscript below. Added text is wavy-underlined and blue, discarded text is struck out and red. There are also minor changes in some figures that are not highlighted in the diff-manuscript below. Additionally, we slightly changed the algorithms and improved the performance. Therefore, some numbers changed in the manuscript.

The reviewer comments are listed below in black. The author's response is written in blue.

**Anonymous Referee #1**

**General comments:**

The manuscript analyses the performance of temperature profile observations by microwave radiometers during rain. This is a relevant topic as more and more of these instruments are operated continuously, and rainy profiles were usually discarded in the past.

I would, however, like to see a more general recommendation on how and under which circumstances these observations can be used. In the discussion, it needs to be clearly stated which channel/angle combination should be used during operation, and which uncertainties are considered to be acceptable, depending on height above ground and rain rate.

We added an additional figure (4) in section 3.2 to highlight which elevation angles and frequencies can be used. Additionally, we added a clear recommendation in the conclusions which retrieval to use.

Concerning the radiative transfer calculations, I would strongly recommend using the updated Rosenkranz gas absorption model from 2022/2023, as this version significantly improves thec radiative transfer results in the lower V-band.

Yes, we agree. We recalculated the radiative transfer with the new Rosenkranz 2022 gas absorption model.

**Specific comments:**

Lines 29ff: Please provide a more precise statement about which "high frequencies" become opaque? The center V-band frequencies (around 58 GHz) are already opaque without rain.

We have specified the statement as suggested.

Lines 46ff: This motivation can be formulated better, such as: "The method presented here can be applied to standard measurement modes and does not require any changes in measurement setup".

Done as suggested.

Lines 63ff: HATPRO doesn't measure voltages, the detectors convert the antenna signal into voltages. A calibration is necessary to convert the voltages into brightness temperatures (but not "the voltages are then calibrated to brightness temperatures")

Changed as suggested to: "For both absorption bands, HATPRO has its own antenna, which measured signal is converted into voltages of the individual frequencies."

Line 69: Add: "during rain"

Done as suggested.

Section 2.2.: Why did you base your retrieval training on old RS80 radiosondes? These sondes are known to have a dry bias, and therefore might also bias your retrieval. I would strongly recommend using state-of-the-art sondes (Vaisala RS41) which have been used in Lindenberg for quite some time already.

Yes, we agree. Our analysis of the radiative transfer calculations are now based on the state-of-the-art sondes RS41. We adapted the script to the new sounding database.

Section 2.4: It seems you did not use all hourly data from 2000 to 2019 for the retrieval development. That would roughly make 20*365*24=175200 time steps, while you used 58195 profiles. How did you select the data? Are the data evenly distributed over the years and seasons?

Yes, you are right. Originally, we used three hourly data, but we changed now to one hour time resolution for our retrieval development. The number of ERA5 profiles is now 173.088. Data range is now from 2004 to 2023.

Section 3, lines 117ff: Please explain better how spectral retrievals work: Individual channels are highly dependent on each other and can thus be used to retrieve the whole spectrum. Did you apply the spectral consistency check only for zenith observations or for all elevation angles? Please comment on that!

We improved the explanation according to your suggestions. Here in this section, the spectral consistency check is only applied to the zenith observations but can also be applied to other elevation angles. In this case *tbx* retrievals for the specific elevation angle need to be created.

Fig. 3: The y-axis range until +/-60 K makes it difficult to see differences between the channels, especially around 0. I would recommend limiting the range to +/-20 K. By the way, I don't think that 10 K bias for some channels in the no-rain case (3a) is satisfactory, whereas in 3b for low elevation angles, the agreement is much better. Do you have an explanation for this? For a bias in the input model data, I would expect that the difference changes with elevation angle.

We completely revised this figure according to your suggestions. We added a second row to see both, absolute values (a, b, c) and their difference (d, e, f). We agree that a 10 K bias for Fig. 3.a) is not satisfactory. However we could not find a specific reason for that. That's why we decided to show the non-cloudy case of another day (27 Jul, 2023), where the bias is less, but unfortunately still present.

Fig.3, continued: Furthermore, I couldn't find a clear explanation of the results in 3c, concerning the different fractions of rain used. Can you discuss that a bit more? Don't you consider the solid lines for 51.26 and 52.28 GHz (yellow, blue) for low elevation angles as significant deviations? Concerning the figure style: I cannot really distinguish the different lines in 3c neither.

During our deeper analysis we found out that our LNM disdrometer observation showed unreasonable values during the rain showers. Especially the drop size distribution did not match to the heavy showers on that day in contrast to disdrometer observations at the TROPOS site 5 km away. Therefore, we decided to remove the LNM disdrometer based size distributions and used only rain rates from the HAPRO weather station to assume size distributions as input fir the PAMTRA simulations.

All in all, the illustration is now easier to interpret and recognise.

Page 8, lines 191ff: This paragraph is a bit confusing, try to make your statements more clearly.

We rephrased the paragraph a bit to make it more clear.

Figure 4: Please provide a somewhat larger plot! (same for Fig. 7)

Our intention was here in the preprint to have the same figure size as later in the two-column final version. We believe that the illustration is easy to read.

Section 4.3: Did you classify the rainy cases using rain rates from the model or from observations?

We used the rain rates from the HATPRO weather station. We added that information.

Figure 8: I would strongly recommend to add a comparative figure (putting the red lines from a,c,e,g and b,d,f,h respectively into one figure). Like this, the additional uncertainties depending on the rain rate can be seen much more clearly. What about the performance of $7v9\varphi$ ? I would be interested to see this combination in comparison here as well.

We changed the figure according to your suggestions.

**Anonymous Referee #2**

I have reviewed the paper titled "Determination of low-level temperature profiles from microwave radiometer observations during rain" by Foth, Lochmann, Saaverdra Garfias and Kalesse-Los. I found the science and its presentation to meet the standards. I hence recommend that this manuscript be accepted with corrections.

**Main questions/ concerns:**

1. L.65-69: discussion on the wet-radome mitigations on the HATPRO radiometer: What is the HATPRO radome made of? Is it still the blue foam? if so, if it gets wet, it will take time to

dry out and, as with sponges, the water will not stop at the top and will fill all the foam by capillarity. This could be problematic for measurements below zenith…

[Figure]

The radome is made of blue foam with an hydrophobic coating. Ageing takes the coating off. We added this information in the manuscript.

2. L.166, L.278 and L.281: On L. 166: "[…] no rain, moderate rain (2.7 mm h−1 ) and heavy rain(3.7 − 11 mm h−1 )". Is moderate defined as 2.7 to 3.6mm/h so that heavy will be defined as 3.7 to 11 mm/h and therefore light will be <2.6 mm/h? your definition is not clear to me. In North America, we have definition for light, moderate and heavy rains: light (< 2.5 mm/h), moderate (2.6 to 7.5 mm/h) and heavy (> 7.6 mm/h); there might be relevant standards in Europe to follow.

Thanks for the hint. We'll apply the definitions from the German weather service:

Light: rain rate < 2.5 mm/h
moderate: rain rate ≥ 2.5 mm/h up to < 10.0 mm/h
heavy: rain rate ≥ 10.0 mm /h
very heavy: rain rate ≥ 50.0/h

Although I agree that the new 4n9j scanning strategy can be used up to 2mm/h (L.278) without too much bias, the following lines (L. 280 – 282) in the conclusion:

"In summary, the HATPRO 4v9φ retrieval method demonstrated in this study achieves unprecedented accuracy of low-level temperature profiling up to 2 km in rain. It was shown that even in heavier rain measurements at elevation angles below 40◦ can be used to derive temperature profiles up to 1.5 km. "

could be interpreted very differently should a casual reader only read the concluding remarks… As such, I would strongly advise that the sentence be re-written much clearer with the limitations of 2mm/h. It is also good practice, even for short papers, to summarize the findings in the conclusion for the casual reader.

Done as suggested.

3. L.250 & Figure 6 : Although I agree that the 4n9j outperforms the other retrievals, there is still some clear influence of the rain in the temperature measurements as seen in Figure 6.h: the retrieved temperature between rain events is smooth like the ECMWF, but during the rain events (3 UTC, 9-13 UTC and 20UTC) the retrieved temperature still shows quite some variability compared to the temperature profiles and still leads to +/- 3K temperature difference. This could be because of the wet radome, as the wet radome emissions are likely angle dependent.

Yes, we agree. This might be caused by rain rates above 2.5 mm h$^{-1}$. We added the discussion in Sec. 4.2.

**Minor fixes:**

- L. 35 – 37: "Xu et al. (2014) retrieved thermodynamic profiles such as temperature and humidity as well as liquid water profiles by using off-zenith MWR observations at 15∘ elevation to reduce the impact of rain on the measurements. As retrieval technique Xu et al. (2014) used a neural network approach. " would be clearer if written:

Xu et al. (2014) retrieved thermodynamic profiles such as temperature and humidity as well as liquid water profiles by using off-zenith MWR observations at 15∘ elevation to reduce the impact of rain on the measurements [using] a neural network approach.

Done as suggested.

- L. 82: "Their accuracy in contrast to other types of radiosondes is described by Turner et al. (2003). " It is unclear to me where this phrase is going. How does the accuracy of the RS80 compare to other types namely the RS41 which is used by the Lindenberg site (MOL-RAO).

We rephrased this section, since we no longer use the RS80.

- L. 91-95: there is not a single reference to ERA5 papers. At the very least the following paper should be referenced:Hersbach, H. et al. 2020: The ERA5 global reanalysis. Q. J. of the R. Meteorol. Soc., 146, 1999–2049.https://doi.org/10.1002/qj.3803.

Thanks. We added the reference.

- L. 118 – 120: "The second panel (b) shows the results of the spectral consistency checks which is retrieved by the so-called tbx retrievals, which work as follows. There are 14 HATPRO frequencies and only 13 of them are used to estimate the expected value for the 14$^{th}$ frequency and then the difference between the estimated and the measured brightness temperature is determined. " would be clearer if written:

The second panel (b) shows the results of the spectral consistency check which is retrieved by the so-called tbx retrieval [. During spectral consistency check (tbx retrievals), 13 of the 14 HATPRO frequencies are used to estimate the value of the unused frequency which is then compared to the measured brightness temperature and the discrepancy is noted.]

Done as suggested.

- Figure caption on Figure 2. "Time series of Moon or Sun and rain quality flag (a), spectral consistency quality flag (b), air temperature and rainfall rate from HATPRO's weather station (c), and height-time series of temperature profiles based HATPRO's firmware radiometer retrieval algorithms in Lindenberg (Germany) on Aug 26, 2020. tb in the colorbar (b) means brightness temperature. "

Should be:

Time series of Moon or Sun and rain quality flag (a), spectral consistency quality flag (b), air temperature and rainfall rate from HATPRO's weather station (c), and height-time series of

temperature profiles based HATPRO's firmware radiometer retrieval algorithms in Lindenberg (Germany) on Aug 26, 2020. tb in the colorbar ( [d] ) means brightness temperature.

Done as suggested.

**Anonymous Referee #3**

The paper is about determining temperature profiles with the help of elevation scanning MWRs during rain and how accurate these profiles are. Usually temperature profile retrievals during rain are not possible due to increased opaqueness of the troposphere within the V-band during rain and due to water accumulation on top of the radome. This paper introduces a method on how to retrieve accurate enough temperature profiles in the lower troposphere in spite of these conditions. Key aspects in doing so are only using off-zenith observations and only utilizing the four optical thickest V-band frequencies.

**General comments**

In my opinion, the necessary quantification of how well the proposed temperature retrieval performs during rainy conditions is missing. Most information is there within the figures but is not stated explicitly in the text.

Overall, the paper is written well and is easy to understand but sometimes details are missing. I will provide more detailed comments and suggestions on what to change below.

We added missing quantification, mentioned in the specific comments below, in the appropriate sections.

General question: What about snowfall? Temperature profiles are usually also not retrieved during snowfall, right? Maybe state in the introduction why you dismiss snowfall and only look into liquid precipitation.

Snow and ice do not emit in the HATPRO frequencies. Cirrus clouds or snow do not disturb the observations. We added the information in the introduction.

**Specific comments**

Abstract: Quantification missing. What's the accuracy of the new retrieval in different rainfall scenarios?

We added the missing quantification.

32: retrieved Ts from NN approach and 1DVar technique? Here the flow of text seems to suggest that the 1DVar is a form of retrieval (NN and MLR), but it isn't, is it? I think it important to tell a little more what the 1DVar is/does or what it means.

A 1DVAR is a one-dimensional variational approach (also known as optimal estimation technique) and can be seen as an assimilation of an observation in an atmospheric state as first guess. One can retrieve atmospheric profiles by this method. Detailed information are provided by the given reference (Ware et al., 2013).

41: reduced by how much? Your method reduces the error during rain EVEN FURTHER? Should make that clear.

At this point, there is no comparison to our method. Araki et al. (2015) used other rain rate bins and compared their results to soundings, whereas we used statistics based on hourly data of ECMWF profiles. We rephrased the sentences a bit to make it clearer.

Maybe cite Böck et al., 2024 in the introduction section. They look into external measurement uncertainties of scanning HATPROs and what these mean for retrieved temperature profiles.

Done as suggested.

45: "almost saturated" with what? is this quantifiable or is there a source? Or is this something you found out in this study?

By this we mean that the signal in these channels is almost saturated in the sense that there are no more extreme jumps due to liquid water, as is the case in the other channels, for example. The difference in transmissivity is one if the basic principles of MWR temperature profiling.

61: better write: "in the order of seconds". I've seen for exact 5min measurements, that there are only ~250 data points and not 300, as expected. So not really a 1s resolution, rather ~1.2s.

Done as suggested.

62: K-band, not Ka-band! Please change this in the whole manuscript. 101-102: There are newer Rosenkranz models. Why did you use an older one? Explain.

Done as suggested.

Would using a newer gas absorption model make a difference? For showing what you want to show, the old model is sufficient I guess.

We applied the newer Rosenkranz 2022 absorption model according to suggestions of referee #1.

118: tbx/SPC Retrievals: Are there more details needed for how 13 frequencies predict the 14th?

No, the tbx retrieval here are only based on measured brightness temperature.

150: Maybe explain shortly why only the upper 4 frequencies for elevation scans are used and the lower 3 frequencies for zenith (à optical thickness)

*The explanation is given in the following section (3.2) where we describe the selection of frequencies and elevation angles in detail.*

166: no range for moderate rain? Why exactly 2.7mm/h?

*This refers to Fig. 3 which we already modified. For details see our comments to referee #1. The number refers to the specific rain events in Fig. 3.*

177-190: I'd wish for a little bit more quantification here; by how much do TBs differ? (It can be seen in the Figure, but it is not written anywhere).

*We added the quantification.*

And what is the threshold for significant difference (when does the pink shaded area start and why?)

*We added an explanation for the significant difference.*

196: "by the less and more transparent..."? get rid of the word less or rephrase.

*Done as suggested.*

Figure3: y-axis title: change it to Delta TB or something similar, to make clear you're talking about a difference of brightness temperatures here. Maybe just call the shade of color pink instead of rose.

*We changed the label of the y-axis.*

200: "degreeS of freedom". Please change in the whole manuscript. Also "gives the information content..." sounds strange. I would rephrase.

*Done as suggested.*

201: You always write "degrees of freedom of signal". Do you always need the word signal or can you omit it?

*We introduced the abbreviation DFS, which makes it easier to read.*

213-222: Can you quantify the differences a little more in the text? In general, you often describe Figures only qualitatively.

*We added more quantification.*

229-239 and Figure5: When talking about bias in this context, wouldn't it be better if you evaluate its variance/accuracy as RSME instead of standard deviation? (same for Fig.8). Or is this bias a mean we're talking about and the spread of this mean is then the SD?

*Bias is the mean of the absolute differences. We replaced the SD by the RMSE as suggested.*

Again: I think it would be better to also quantify your findings in the text.

*Yes, we agree and we added more quantifications*

240-250: Again: quantify also in the text.

Yes, we agree and we added more quantifications

251-260: Here you do quantify the differences in the text. You should do that everywhere.

Yes, we agree.

Figure8: Again: In this context I'm not sure if you should rather talk about RSME instead of standard deviation. You should check that.
The values won't change much, as the only difference is that you divide by n and not n−1.

It is now the RMSE.

Why is the bias of the 4vz10phi that much worse above 1km in the no rain scenario?

One has to keep in mind that we do not compare here to the truth values (reality), but only to ECMWF model data, which might also be biased. Therefore, the focus here is more on the relative difference between the retrievals and not so much on the absolute values.

276-279: Quantify: How much better does the new 4v9phi retrieval perform and/or with what accuracy during rainfall up to 2mm/h? You only quantify the case in the text with a rainrate of below 0.5mm/h.

We added more quantifications in the manuscript.

280-end: Quantification is missing in the conclusion.

How much Kelvin exactly is the new retrieval method better compared to the standard one?

See comment after next.

E.g. for slight rain below 2km: How much Kelvin is this different to non-rain conditions?

See next comment.

And how much is it different below 1.5km for heavy rain?

A comparison with the standard retrieval during rain is meaningless, as the default retrieval is not intended to be used during rain. The default retrieval is expected to give unreliable results during rain for the reasons mentioned in Sect. 1 and would never be used for that purpose. Temperature profiling during rain is the novelty of our presented approach.

We added more quantification,

In general: What's the general temperature profile accuracy of elevation scanning state-of-the-art HATPROs for no rain scenarios and how does it compare to the new findings?

We added more quantification.

I think this is important for the reader, so they can better classify/categorize the results of this paper.

We agree.

[revised manuscript text omitted]

---

## Author Response (AR2)

**Response to Reviewers #1 and #2**

We like to thank the reviewers for providing helpful comments to improve the manuscript.

All changes are highlighted in the diff-manuscript below. Added text is wavy-underlined and blue, discarded text is struck out and red. The reviewer comments are listed below in black. The author's response is written in blue.

**Anonymous Referee #1**

**General comments:**

The revised manuscript of "Determination of low-level temperature profiles from microwave radiometer observations during rain" incorporated all major comments and concerns from the first review round.

The revised manuscript is now all right content-wise but sometimes suffers from some minor language difficulties.

**Some overall comments:**

You mention brightness temperatures a lot. You could think about introducing Tb as an abbreviation. Same goes for temperature profile. Consider using T-profile.

We agree. Done as suggested.

You could make it a little more clear (probably in the summary as well) that when a radome gets wet everywhere (not only at the top) due to old age, that none of your proposed retrievals will work properly during (and shortly after) rain events and that monitoring/knowing the state of the radome is very important. Your method relies heavily on a radome in pristine condition. If the radome hasn't been replaced for a long time, is damaged and is not hydrophobic anymore, the accuracy of your T-profile retrievals during rain will suffer.

We agree and added the clarification as suggested in the microwave radiometer HATPRO section 2.1.

**Some minor corrections in detail:**

line 8-10: We found out,/We show,/ It is shown, that retrievals... provide the best results.

Done as suggested.

line 44: ...with a 1DVAR technique…

Done as suggested.

line 78: multi-variate linear regressions (I would omit the word model here)

Done as suggested.

line 109: ...and was, e.g., also applied in Löhnert and Crewell (2003), Löhnert et al. (2007), and Foth and Pospichal (2017).

Done as suggested.

line 120f: ...retrievals during rain, indicated by unrealistic spikes, is shown…

Done as suggested.

...the rain and sun quality flags (a) denote if... (or is it actually one and the same flag?)

Done as suggested.

line 125: ...During spectral consistency checks. OR ...During a spectral consistency check.

Done as suggested.

line 128: ...for the K-band ...for the V-band.

Done as suggested.

line 130: ...have passed…

Done as suggested.

line 141f: A (temperature) retrieval....multi-variate linear regression method... (I wouldn't talk about a model here, see line 78)

Done as suggested.

line 142f: In this work we use the regression method.

Done as suggested.

line 152: regression method

Done as suggested.

line 159: ...are used at only the zenith angle.

Done as suggested.

line 181: ...was chosen…

Done as suggested.

line 187: Additionally, horizontally homogeneous atmospheric conditions are assumed. /a horizontally homogeneous atmosphere is assumed.

Done as suggested.

I think you don't always need to write the whole term "elevation angle". Usually it's enough to talk about, e.g., 30° elevation. Omitting the word angle often makes long sentences easier to read.

But refrain from just using the word angle alone. Replace angle with elevation (e.g. at end of line 202 and line 227).

We understand the concern but we don't agree here. We think it is more precise to use the full term.

line 217f: You can delete: "Horizontally homogeneous conditions are assumed for boundary layer scans". You already mentioned that before.

Done as suggested.

line 241: ...retrievals show…

Done as suggested.

line 255: ...the prediction from the regression method…

Done as suggested.

line 309: To summarize, the…

Done as suggested.

line 315: ...up to 3 km during rain (events) with rain rates…

Done as suggested.

lines 323ff: I think mentioning the different retrievals only with their abbreviation in the text sounds a little odd.

We do not believe that it would make sense to stop using the abbreviations in the summary.

line 331ff: Rephrase. Something's wrong here in the sentence structure.

Done as suggested.

line 336: ...to increase the performance of the... (even) further…

Done as suggested.

**Anonymous Referee #1**

**General comments:**

The manuscript has much improved during the review process, therefore I suggest that it should be published after some small minor revisions.

**Detailed comments:**

- In section 3.1/3.2: Please make it clear that the retrievals you created (4v9f, etc.) are not based on PAMTRA simulations, but on the non-scattering RT model. This might be confusing to some readers!

This is already done in Sec. 3.1 with the following sentence "This information is used as input to the non-scattering microwave radiative transfer model (see Sec.2.5). "

- lines 23-24: "Snow and ice clouds do not emit in the considered spectrum, hence they are not taken into account here". This phrase is too early here, as you don't talk about MW observations yet, but only very generally about remote sensing.

We have moved the sentence a little further back in the introduction.

- lines 27-28: "Additionally, radiosondes show a significant sonde-to-sonde variability (Nash et al., 2005) as well as a dry bias (Turner et al., 2003)." This is less relevant for the recent RS41 soundings, these papers refer to older models (e.g. Vaisala RS80). I would omit it therefore.

We omit the sentence.

- lines 34-35: You write: "During rain the atmosphere becomes opaque at high frequencies of the V-band (...)". This is not correct - also during non-rainy conditions, the atmosphere is opaque at these frequencies! Please correct/rephrase!

We omit the sentence and rephrased the next sentence a bit.

- lines 47-49: please check this phrase again, I think there are some words missing.

We rephrased the sentence a bit.

- Section 4.1 and conclusions: Shouldn't it be "7v9f"? You write "7v10f" several times in this section. Please check throughout the whole document!

Yes you are right. Done as suggested.